🔓 | **Open Peer Review** | Immunology | Research Article

# Analysis of SARS-CoV-2 isolates, namely the Wuhan strain, Delta variant, and Omicron variant, identifies differential immune profiles

Shima Shahbaz,[1] Najmeh Bozorgmehr,[1] Julia Lu,[1] Mohammed Osman,[2] Wendy Sligl,[3,4] D. Lorne Tyrrell,[5,6] Shokrollah Elahi[1,6,7,8,9]

**ABSTRACT**   There is an urgent need to better understand the impact of different severe acute respiratory syndrome coronavirus 2 (SARS-CoV-2) variants on immune response and disease dynamics to facilitate better intervention strategies. Here, we show that SARS-CoV-2 variants differentially affect host immune responses. The magnitude and quantity of cytokines and chemokines were comparable in those infected with the Wuhan strain and the Delta variant. However, individuals infected with the Omicron variant had significantly lower levels of these mediators. We also found an elevation of plasma galectins (Gal-3, Gal-8, and Gal-9) in infected individuals, in particular, in those with the original strain. Soluble galectins exert a proinflammatory role in COVID-19 pathogenesis. This was illustrated by their correlation with the plasma levels of sCD14, sCD163, enhanced TNF-α/IL-6 secretion, and increased SARS-CoV-2 infectivity *in vitro*. Moreover, we observed enhanced CD4+ and CD8+ T cell activation in Wuhan strain-infected individuals. Surprisingly, there was a more pronounced T cell activation in those infected with the Omicron in comparison to the Delta variant. In line with T cell activation status, we observed a more pronounced expansion of T cells expressing different co-inhibitory receptors in patients infected with the Wuhan strain, followed by the Omicron and Delta variants. Individuals infected with the Wuhan strain or the Omicron variant had a similar pattern of plasma soluble immune checkpoints. Our results imply that a milder innate immune response might be beneficial and protective in those infected with the Omicron variant. Our results provide a novel insight into the differential impact of SARS-CoV-2 variants on host immunity.

**IMPORTANCE**   There is a need to better understand how different SARS-CoV-2 variants influence the immune system and disease dynamics to facilitate the development of better vaccines and therapies. We compared immune responses in 140 SARS-CoV-2-infected individuals with the Wuhan strain, the Delta variant, or the Omicron variant. All these patients were admitted to the intensive care unit and were SARS-CoV-2 vaccination naïve. We found that SARS-CoV-2 variants differentially affect the host immune response. This was done by measuring soluble biomarkers in their plasma and examining different immune cells. Overall, we found that the magnitude of cytokine storm in individuals infected with the Wuhan strain or the Delta variant was greater than in those infected with the Omicron variant. In light of enhanced cytokine release syndrome in individuals infected with the Wuhan strain or the Delta variant, we believe that a milder innate immune response might be beneficial and protective in those infected with the Omicron variant.

**KEYWORDS**   COVID-19, co-inhibitory receptors, galectins, cytokines, soluble immune checkpoints

Address correspondence to Shokrollah Elahi, elahi@ualberta.ca.

The authors declare no conflict of interest.

See the funding table on p. 21.

The emergence of severe acute respiratory syndrome coronavirus 2 (SARS-CoV-2) has resulted in a global pandemic with devastating outcomes. Although most infected individuals present with mild to moderate symptoms, some result in life-threatening respiratory infections with diverse complications. As of July 2023, the COVID-19 pandemic had resulted in 6.8 million deaths among 690 million infected cases, which accounts for ~1% fatality rate. Among infected individuals, about 3–20% require hospitalization, of which 10–30% need intensive care (1). As the pandemic continues, so does the evolution of SARS-CoV-2. As a result, several new variants of concern (VoC) such as the Alpha, Beta, Gamma, Delta, and Omicron have emerged. Variants such as Delta and Omicron remained as dominant circulating SARS-CoV-2 strains due to their enhanced transmissibility and immune evasion (2, 3). For example, the Omicron variant has been associated with lower rates of pneumonia and disease severity in intensive care unit (ICU)-admitted patients and fewer in-hospital deaths than the Delta variant (4). Moreover, in a large cohort of infected individuals with the Alpha, Gamma, Delta, and Omicron variants, the disease severity appeared to be the same for the three former variants but less severe for the latter (5). However, it is not well defined whether SARS-CoV-2 variants exert differential effects on immune and non-immune cells.

Considering the diverse mutations in SARS-CoV-2 variants, it is possible that innate immune cells respond differently to these variants through pattern recognition receptors or damage recognition receptors. For instance, activated macrophages/monocytes, neutrophils, and natural killer (NK) cells can amplify the "cytokine release syndrome" (6), which has been associated with a higher mortality risk in COVID-19 patients (7).

It has been shown that antigen-specific CD8$^+$ T cells play an essential role in clearing virus-infected cells (8, 9, 10). Likewise, CD4$^+$ T cells promote the magnitude of CD8$^+$ T cell response and enhance their clonal expansion and differentiation (10–12). Activated T cells recruited to the infection site (e.g., lungs) may promote T cell-dependent cytokine release and cytotoxicity, further augmenting the pathogenesis (13). Recent evidence suggests that the latter may contribute to the disease's severity (14). As such, transient upregulation of co-inhibitory receptors (e.g., immune checkpoints) is important for minimizing T cell-mediated immunopathology (15). Although earlier COVID-19-related studies suggested that the upregulation of co-inhibitory receptors (TIM-3, NKG2D, and PD-1) was associated with T cell exhaustion (16, 17), this concept was later challenged based on the observations that SARS-CoV-2-specific T cells exhibited an activated but not impaired phenotype in infected individuals with the original strain (18). In fact, overexpression of co-inhibitory receptors is reflective of T cell activation as illustrated by the expansion of CD38$^+$, HLA-DR$^+$, and Ki67$^+$ CD8$^+$ T cells in the peripheral blood of infected individuals with the Wuhan strain (18–20).

In addition to the cell membrane-bound immune checkpoints, soluble immune checkpoints (sICs) are released by the cleavage of membrane-bound protein or by enhanced mRNA expression/production (21). These have been correlated with disease severity in COVID-19 patients (22, 23). Beyond cytokines, chemokines, and sICs, soluble CD14 (sCD14) and soluble CD163 (sCD163) as myeloid differentiation and activation markers are elevated in the plasma of COVID-19 patients and associated with immunopathology (24). It has also been demonstrated that different types of galectins (Gals) including Gal-1, Gal-3, Gal-8, and Gal-9 were elevated in the plasma of individuals infected with the Wuhan strain (6, 25–27). Gals are widely expressed in immune and non-immune cells and bind to carbohydrates and proteins (28). They are involved in several biological processes such as development, signal transduction, and immune responses (8, 28–30). Although the pathogenesis of the original strain of SARS-CoV-2 has been extensively studied, there is very limited information regarding its close relatives, Delta and Omicron. Therefore, we sought to investigate whether the original strain of SARS-CoV-2 and its Delta and Omicron variants differentially affect immune responses in ICU-admitted patients. In this study, we quantified and compared soluble analytes including 30 different cytokines/chemokines, sCD14, sCD163, Gal-3, Gal-8, Gal-9, and sICs in the plasma of 140 SARS-CoV-2-infected and ICU-admitted individuals with the

Wuhan strain and the Delta/Omicron variants. Moreover, we assessed T cell activation and measured the frequency of CD4$^+$ and CD8$^+$ T cells expressing different co-inhibitory/co-stimulatory molecules in our cohorts compared to 25 uninfected controls.

## MATERIALS AND METHODS

### Study population

Blood samples for this study were collected from 140 ICU-admitted COVID-19 patients in Edmonton, Canada, including those infected with the Wuhan strain ($n = 46$, age 56 ± 11), the Delta variant ($n = 47$, age 53 ± 14), and the Omicron variant ($n = 47$, age 56 ± 13). We also recruited healthy controls (HCs) ($n = 25$, age 50 ± 12) for comparison. Patients were SARS-CoV-2 positive by reverse transcription-polymerase chain reaction (RT-PCR) assay specific for viral RNA-dependent RNA polymerase using a nasopharyngeal swab or endotracheal aspirates. The strain/variant-specific PCR was not performed on the entire cohort, but it was performed randomly. Based on random PCR and the timing of the diagnosis relative to dominant strains at the time, we have little doubt in regard to the strain/variant determination. To prevent the interference of vaccination against SARS-CoV-2 with immune responses to infection, we only recruited unvaccinated patients for these studies. Of note, our patients did not have a previous symptomatic infection with another SARS-CoV-2 strain/variant considering their differential emergence time.

### Sample collection and processing

Fresh peripheral blood was collected in EDTA tubes and processed for plasma collection. Then, fresh peripheral blood mononuclear cells (PBMCs) were isolated over Ficoll (GE) gradient, and cell cultures were performed in RPMI 1640 (Sigma-Aldrich) supplemented with 10% fetal bovine serum (FBS) (Sigma-Aldrich) and 1% penicillin/streptomycin (Sigma-Aldrich).

### Cell culture and flow cytometry

The fluorochrome-conjugated antibodies were purchased from ThermoFisher Scientific, BD Biosciences, or BioLegend. Specifically, the following antibodies were used: anti-CD3 (HIT3a), anti-CD4 (RPA-T4), anti-CD8 (RPA-T8), anti–TIM-3 (7D3), anti–PD-1 (MIH4), anti-CD160 (BY55), anti-CD244 (DM244), anti–galectin-9 (Gal-9) (9M1-3), anti-TIGIT (MBSA43), anti-CD39 (TU66), anti-CD73 (AD2), anti-CD26 (M-A261), anti-CD38 (HIT2), anti–HLA-DR (LN3), anti-CD71 (MA712), anti–TNF-α (MAB11), and anti–IFN-γ (4S.B3). The LIVE/DEAD Fixable Dead Cell Stains (Thermo Fisher Scientific) were used for the exclusion of dead cells. In some experiments, PBMCs from HCs were cultured with or without recombinant human Gal-3 (rhGal-3; Abcam, 0.02 µg/mL), recombinant human Gal-8 (0.02 µg/mL) (kindly provided by Dr. Joanne Lemieux, University of Alberta), or recombinant human Gal-9 (rhGal-9; Gal-Pharma, Japan, 0.02 µg/mL), in the presence or absence of lipopolysaccharide (LPS, 1 µg/mL, Sigma-Aldrich) plus the protein transport inhibitor brefeldin-A (BD Biosciences) for 6 h for intracytoplasmic staining and without protein transport inhibitor brefeldin-A overnight for enzyme-linked immunosorbent assay (ELISA). Next, treated cells were subjected to surface staining for T cell or monocyte markers depending on the experiment followed by fixation and permeabilization (BD Cytofix/Cytoperm) and intracellular cytokine staining (ICS) for TNF-α and IFN-γ staining according to our protocols and previous protocols (18, 31, 32). Finally, fixed cells in paraformaldehyde (4%) were acquired on flow cytometry using a Fortessa-X20 or LSR Fortessa-SORP flow cytometer (BD Biosciences) and analyzed with FlowJo software (version 10).

### Multiplex cytokines, chemokines, and sICs measurements

Plasma samples were centrifuged for 15 min at 2,000 × $g$ and diluted at two- and fourfold for cytokines and chemokines analyses, respectively. Cytokines and chemokines were

measured using the V-PLEX proinflammatory panel 1, cytokine panel 1, and chemokine panel 1 kits from Meso Scale Discovery (MSD) according to the manufacturer's instructions and our reports (6, 33). The following analytes were quantified: IL-6, TNF-α, IL-10, IFN-γ, IL-13, IL-1β, IL-2, IL-12p70, IL-4, GM-CSF, IL-1α, IL-5, IL-7, IL-12/23p40, IL-15, IL-16, IL-17, VEGF-A, TNF-β, Eotaxin, MIP-1α, Eotaxin-3, IL-8, TARC, IP-10, MIP-1β, MCP-1, MDC, and MCP-4. A total of 65 plasma samples from COVID-19 patients and 15 plasma samples from HCs were examined for these studies. In addition, the concentration of immune checkpoints was measured using the U-PLEX effector cell checkpoint combo 1 kit from MSD following the manufacturer's instruction. The concentration of the following molecules was measured: CD27, CD28, CD40L, TIM-3, LAG-3, OX40, PD-1, TIGIT, GITR, and CTLA-4. Data were acquired on the V-plex Sector Imager 2400 plate reader, and the concentration of analytes was calculated using the MSD Workbench 3.0 software. For other ELISAs, the following kits from R&D were used: Gal-3 (DY1154), Gal-9 (DY2045), soluble CD14 (DC140), soluble CD163 (DY1607-05), IL-18 (DL180), IL-6 (DY206-05), and TNF-α (DY210). The plasma Gal-8 was measured using the human Gal-8 ELISA kit from Thermo Fisher Scientific (EH205RB).

## Infection assay with the pseudo-SARS-CoV-2

Vero E6 cells were seeded at 100,000 cells/well in culture media in the presence or absence of Gal-3, Gal-8, and Gal-9 (1 µg/mL or 2 µg/mL) in a 24-well flat-bottomed plate for 24 h. Then, cells were washed with warm phosphate-buffered saline (PBS) and exposed to the pseudo-SARS-CoV-2 spike Delta variant green reporter (Montana Molecular, MT, USA) according to the manufacturer's instruction and our protocol overnight (34). Then, cells were extensively washed of any extracellular pseudovirus and trypsinized before being subjected to flow cytometry analysis.

## Statistical analysis

For statistical analysis, we first determined the distribution of data using the Wilks-Shapiro test, and then, based on the distribution of data, the appropriate test was used. When data were not normally distributed, non-parametric tests such as the Mann-Whitney U-test or Kruskal-Wallis one-way analysis of variance were used. Data are presented as means and standard deviations (mean ± SD), and $P < 0.05$ was considered to be statistically significant. Statistical analysis was performed using GraphPad Prism 9 (GraphPad Software, Inc.). Heap map plots were generated using the R scripts.

## RESULTS

### Patient demographics and comorbidities

Patients' demographics and comorbidities are described in Table S1. Mortality was significantly higher in men versus women, and similar to previous reports, we observed comorbidities such as asthma, type 2 diabetes, cardiovascular diseases, and obesity in our patients. In agreement with other reports, we noted a higher mortality and disease severity in individuals infected with the Wuhan strain or the Delta variant compared to the Omicron variant (Table S1). HCs were age and sex matched with a ratio of 60% males to mimic COVID-19 patients. All patients were diagnosed SARS-CoV-2 positive by quantitative RT-PCR assay specific for viral RNA-dependent RNA polymerase and envelope transcripts using a nasopharyngeal swab. Since the kinetics, phenotype, and T cell function change dramatically over time, we performed our studies on specimens collected from COVID-19 patients 2 weeks post-onset of symptoms and/or SARS-CoV-2 diagnosis.

## Differential levels of cytokines and chemokines in the plasma of SARS-CoV-2-infected individuals with the Wuhan strain and the Delta/Omicron variants

Since the role of cytokines and chemokines in the pathogenesis of SARS-CoV-2 infection and the overall survival of COVID-19 patients has been well documented (6, 7), we decided to quantify a wide range of cytokines and chemokines in the plasma of SARS-CoV-2-infected individuals with the original strain versus the Delta and Omicron variants.

Principal component analysis (PCA) based on Euclidean distances separated HCs from COVID-19 patients in a two-dimensional plot (Fig. 1A). Although HCs were clustered together, SARS-CoV-2-infected individuals did not show a distinct separation from each other. It appeared that those infected with the Wuhan strain were mostly clustered together except for two patients who deployed a cytokine and chemokine profile similar to that of HCs and Delta/micron variants (Fig. 1A). More specifically, compared to HCs, patients infected with both the Wuhan strain and the Delta variant showed significant elevation of IL-6, TNF-α, IL-10, IFN-γ, IL-4, IL-15, Eotaxin, Eotaxin-3, IL-8, IP-10, MIP-1α, MCP-1, MIP-1β, IL-18, vascular endothelial growth factor (VEGF-A), IL-17, and GM-CSF, but a significant reduction in thymus- and activation-regulated chemokine (TARC) and IL-13 in their plasma (Fig. 1B through D; Fig. S1A and B). Moreover, patients infected with the Wuhan stain and Delta variant exhibited higher levels of IL-2 and IL-1β in their plasma, respectively (Fig. 1B through D; Fig. S1A and B). When plasma cytokine and chemokine levels in patients infected with the Omicron variant were compared to HCs, the observed changes were less dramatic compared to the Wuhan strain and the Delta variant. Omicron-infected individuals showed a significant increase in IL-6, TNF-α, IL-10, IFN-γ, IL-4, IL-15, Eotaxin, Eotaxin-3, IL-8, IP-10, MIP-1α, MCP-1, MIP-1β, and IL-18, but a reduction in TARC, MCP-4, and IL-7 levels in their plasma compared to the HC cohort (Fig. 1B through D; Fig. S1C).

A volcano plot of normalized data confirmed similar observations that patients infected with either the Wuhan strain or the Delta and Omicron variants had elevated concentrations of multiple cytokines and chemokines compared to HCs (Fig. 1E through G). In particular, IL-6, IP-10, and IL-10 were more pronounced in infected individuals regardless of the viral strain/variant (Fig. 1E through G). Overall, the numbers of elevated cytokines and their magnitude were less striking in patients infected with the Omicron variant in comparison to the Wuhan strain and Delta variant (Fig. 1E through G).

Next, we compared differences in the plasma cytokines and chemokines levels in our three SARS-CoV-2-infected cohorts. Interestingly, we found no significant difference in the levels of cytokines and chemokines in patients infected with the Wuhan strain and the Delta variant (Fig. 1B through D; Fig. S1D). However, we observed significantly lower levels of IL-18, VEGF-A, IL-7, IP-10, MCP-1, MIP-1α, Eotaxin, IFN-γ, IL-8, TNF-α, and IL-1β in patients infected with the Omicron variant when compared to the Wuhan strain (Fig. 1B through D and 1G; Fig. S1E). When we compared the levels of cytokines/chemokines between the Omicron and Delta variants, we noted a significant reduction in VEGF-A, IL-7, IP-10, MCP-1, Eotaxin, Eotaxin-3, IFN-γ, and IL-1β in those infected with the Omicron (Fig. 1B through D and I; Fig. S1F). The volcano plot of the normalized data also indicated that overall patients infected with the Omicron variant had a reduced inflammatory response compared with patients infected with the ancestral strain and the Delta variant (Fig. 1H and I). In particular, the levels of IP-10 and IL-18 were profoundly lower in the plasma of Omicron-infected patients compared with the Wuhan strain, and the levels of IP-10 and Eotaxin-3, in the plasma of patients infected with the Omicron versus the Delta variant (Fig. 1H and I). Taken together, these observations imply that although a wide range of cytokines and chemokines are elevated in the plasma of SARS-CoV-2-infected individuals, those infected with the Omicron variant exhibit significantly lower levels of plasma cytokines and chemokines compared with those infected with the Wuhan strain or the Delta variant.

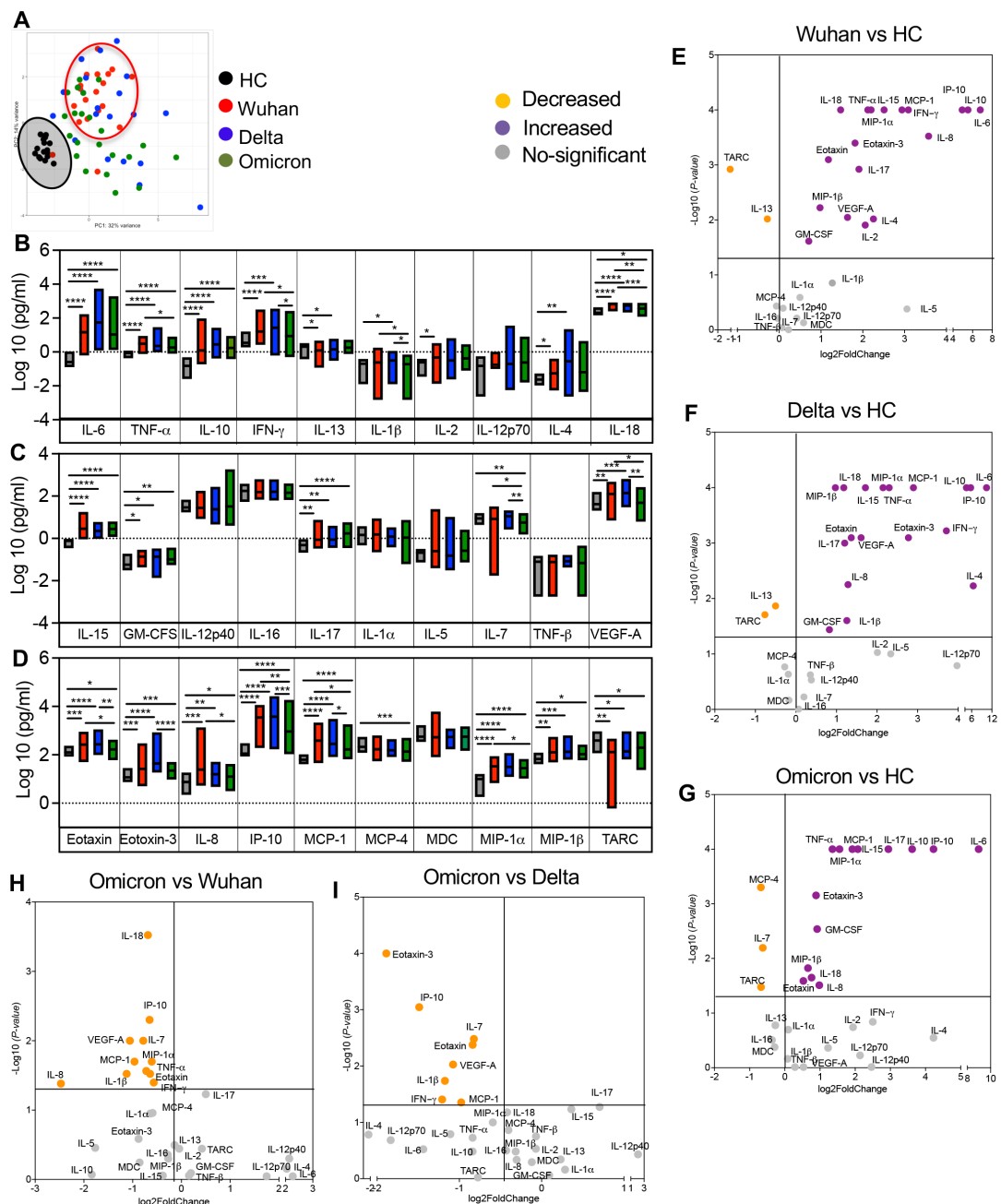

**FIG 1** The elevation of a wide range of cytokines and chemokines in the plasma of COVID-19 patients. (A) The PCA on the Euclidian distances between cytokine and chemokine profiles of different study cohorts. (B to D) Normalized and calculated concentrations of cytokines and chemokines in the plasma of COVID-19 patients infected with different strains ($n = 62$) vs HCs ($n = 16$) as measured by MSD. The volcano plot illustrating the magnitude and significance of differences in cytokines/chemokines concentrations in individuals infected with SARS-CoV-2: (E) Wuhan vs HC, (F) Delta vs HC, (G) Omicron vs HC, (H) Omicron vs Wuhan, and (I) Omicron vs Delta variants. Circles marked in purple and orange are significantly increased or decreased, respectively, but circles in gray indicate no significant difference. The significant difference in median plasma concentration was calculated for multiple testing. *, $P < 0.05$; **, $P < 0.01$; ***, $P < 0.001$; ****, $P < 0.0001$.

## The original strain of the SARS-CoV-2 and its variants, namely Delta and Omicron, are associated with the induction of different levels of plasma Gals, sCD14, and sCD163

It has been reported that plasma Gal-3 and Gal-9 levels were elevated with profound proinflammatory properties in those infected with the Wuhan strain (6, 27, 35). However, the effects of different SARS-CoV-2 variants on soluble Gal levels in the plasma of

infected individuals have not been investigated. Therefore, we investigated whether Gal-3, Gal-8, and Gal-9 levels differed in the plasma of infected individuals with the Wuhan strain versus the Delta and Omicron variants. Interestingly, we found that the plasma concentrations of all three Gals were significantly elevated in the plasma of individuals infected with the Wuhan strain and the Delta variant compared with HCs (Fig. 2A through C). However, this was not the case for individuals infected with the Omicron variant, as this cohort showed only the elevation of plasma Gal-9 but not Gal-3 and Gal-8 compared with HCs (Fig. 2A through C). Moreover, our results revealed that patients infected with the Omicron variant had significantly lower levels of Gal-3, Gal-8, and Gal-9 compared with those infected with the original strain of SARS-CoV-2 (Fig. 2A through C). Of note, infected individuals with the Delta variant had comparable levels of Gal-3 and Gal-9 but significantly lower plasma concentrations of Gal-8 compared with those infected with the Wuhan strain (Fig. 2A through C). Finally, we observed that Omicron-infected patients had significantly lower levels of Gal-3 and Gal-9 but comparable levels of Gal-8 in comparison to individuals infected with the Delta variant (Fig. 2A through C). Overall, our observations demonstrate that the levels of Gals in the plasma of infected patients were more pronounced in those infected with the Wuhan strain, followed by the Delta variant and the Omicron variant.

Considering the immunomodulatory roles of Gals (28) and their potential roles in cytokine storm in COVID-19 patients (6, 27), we examined the correlation of Gal-3, Gal-8, and Gal-9 levels with detected cytokines and chemokines in the plasma of our cohorts. These analyses revealed a positive and significant correlation between the levels of plasma Gal-9 with MIP-1β, MIP-1α, MCP-1, TNF-α, IP-10 in individuals infected with the Wuhan strain (Fig. 2D through H). Surprisingly, we noted a positive correlation for MIP-1β and IL-18 with Gal-9 levels in plasma samples from individuals infected with the Delta variant and only IL-18 in those infected with the Omicron variant (Fig. 2I through K). A similar analysis showed a positive and significant correlation between Gal-8 concentrations with Eotaxin, IL-6, and TNF-α levels in the plasma of individuals infected with the Wuhan strain (Fig. 2L through N). Only IL-6 was found to be correlated in individuals infected with the Delta variant (Fig. 2O), but no correlation was observed between Gal-8 levels with cytokines/chemokines in individuals infected with the Omicron.

Finally, we analyzed the correlation of plasma Gal-3 with detected cytokines/chemokines in our cohorts, which revealed a strong correlation between the plasma Gal-3 concentrations with IL-6, TNF-α, GMC-SF, MDC, and IL-8 in individuals infected with the Wuhan strain (Fig. 3A through E). A similar positive correlation was observed for the plasma Gal-3 with IL-6, TNF-α, and MCP-1 in individuals infected with the Delta variant (Fig. 3F through H). However, Gal-3 levels did not show any correlation with detected cytokines/chemokines in the plasma of individuals infected with the Omicron variant.

As proof of concept to validate our correlation studies, we treated PBMCs from HCs with physiological concentrations of Gal-3, Gal-8, and Gal-9 for 16 h to assess their effects on cytokine production. We found that all three Gals significantly enhanced TNF-α and IL-6 production by PBMCs, but it was more pronounced for Gal-3 (Fig. 3I and J). Moreover, we observed that Gal-3 and Gal-8 but not Gal-9 enhanced TNF-α and IL-6 expression in monocytes after 6-h culture (Fig. 3K through M). In particular, we noted a synergistic effect when PBMCs were treated with all three Gals (Fig. 3K through M).

Moreover, we quantified sCD14 and sCD163 levels in the plasma as two inflammatory markers associated with monocyte/macrophage activation (36). There are reports on the elevation of plasma sCD14 and sCD163 levels in SARS-CoV-2-infected individuals with the Wuhan strain, which is associated with the disease severity (24). We observed that sCD14 and sCD163 levels were significantly elevated in the plasma of patients from all three infected cohorts when compared to HCs (Fig. 4A and B). Of note, Delta-infected patients exhibited significantly lower levels of sCD163 in their plasma compared with those infected with the Wuhan strain but not the Omicron variant (Fig. 4B). Next, we analyzed the correlation of sCD14 and sCD163 with Gals in the plasma of different cohorts. We noted a positive and significant correlation between the plasma Gal-9 levels

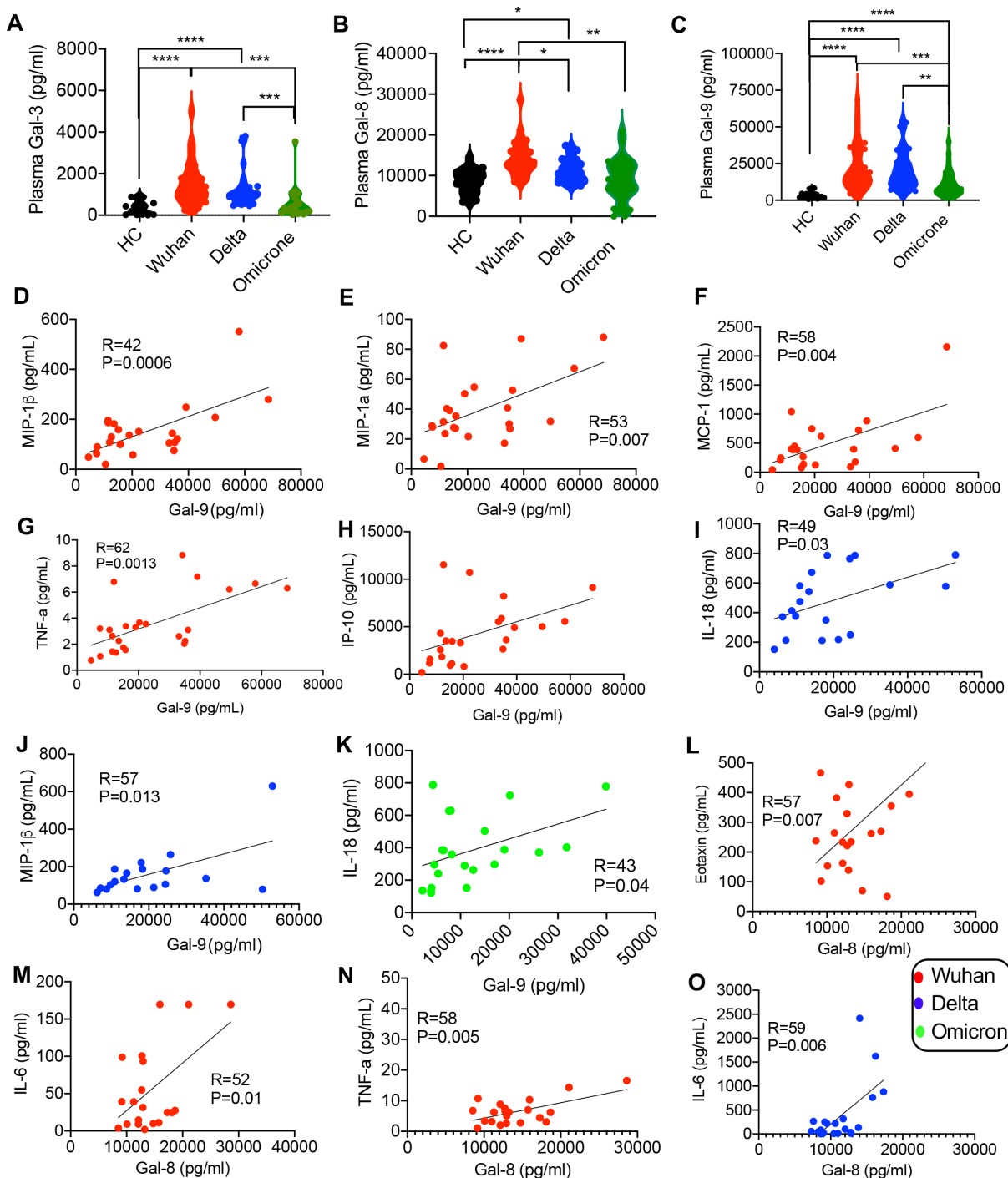

**FIG 2** Differential elevation of plasma Gal-3, Gal-8, and Gal-9 in COVID-19 patients infected with the Wuhan strain vs Delta and Omicron variants. Detected concentrations of (A) Gal-3, (B) Gal-8, and (C) Gal-9 in the plasma specimens of different cohorts as measured by ELISA. (D) Correlation of the plasma MIP-1β with Gal-9 concentrations in the Wuhan cohort. (E) Correlation of MIP-1α with Gal-9 in the Wuhan cohort. (F) Correlation of MCP-1 with Gal-9 in the Wuhan cohort. (G) Correlation of TNF-α with Gal-9 in the Wuhan cohort. (H) Correlation of IP-10 with Gal-9 in the Wuhan cohort. (I) Correlation of IL-18 with Gal-9 in the Delta cohort. (J) Correlation of MIP-1β with Gal-9 in the Delta cohort. (K) Correlation of IL-18 with Gal-9 in the Omicron cohort. (L) Correlation of plasma Eotaxin levels with Gal-8 in the Wuhan cohort. (M) Correlation of IL-6 levels with Gal-8 in the Wuhan cohort. (N) Correlation of TNF-α levels with Gal-8 in the Wuhan cohort. (O) Correlation of IL-6 with Gal-8 levels in the Omicron cohort.

but not the other two Gals with sCD14 in all three COVID-19 cohorts (Fig. 4C through E). However, this was not the case for sCD163 when analyzed in all cohorts.

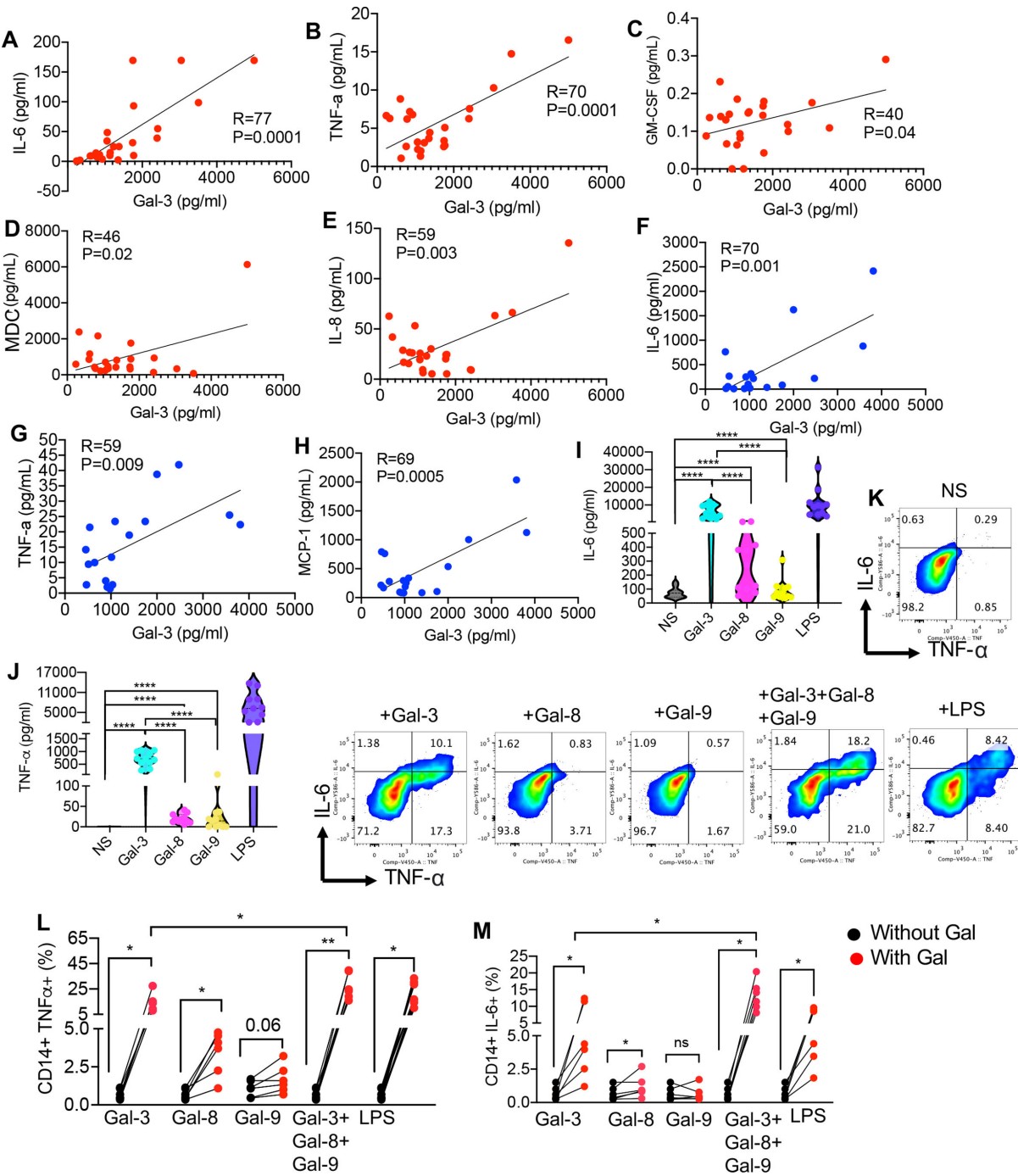

**FIG 3** Gal-3 displays a more potent proinflammatory profile than Gal-8 and Gal-9. (A) Correlation of IL-6 with Gal-3 levels in the plasma of the Wuhan cohort. (B) Correlation of TNF-α with Gal-3 in the Wuhan cohort. (C) Correlation of GM-CSF with Gal-9 in the Wuhan cohort. (D) Correlation of MDC with Gal-3 in the Wuhan cohort. (E) Correlation of IL-8 with Gal-3 in the Wuhan cohort. (F) Correlation of IL-6 with Gal-3 levels in the Delta cohort. (G) Correlation of TNF-α with Gal-3 in the Delta cohort. (H) Correlation of MCP-1 with Gal-3 in the Delta cohort. (I) Detected concentrations of IL-6 in culture supernatants of treated PBMCs from SARS-CoV-2 uninfected individuals with rhGal-3 (0.02 µg/mL), rhGal-8 (0.02 µg/mL), and rhGal-9 (0.02 µg/mL) for overnight as quantified by ELISA. (J) Detected concentrations of TNF-α in culture supernatants of treated PBMCs from SARS-CoV-2-uninfected individuals with rhGal-3 (0.02 µg/mL), rhGal-8 (0.02 µg/mL), and rhGal-9 (0.02 µg/mL) for overnight as quantified by ELISA. (K) Representative flow cytometry plots and cumulative data of percentages of (L) TNF-α and (M) IL-6 expressing monocytes (CD14) in treated PBMCs from SARS-CoV-2-uninfected individuals following treatment with rhGal-3 (0.02 µg/mL), rhGal-8 (0.02 µg/mL), and rhGal-9 (0.02 µg/mL) and their combination for 6 h as measured by ICS. The LPS (2 µg/mL) was used as a positive control. *, $P < 0.05$; **, $P < 0.01$; ***, $P < 0.001$; ****, $P < 0.0001$.

To better demonstrate the difference in Gals and sCD14 and sCD163 levels in the plasma of our cohorts, PCA on Euclidean distances was used to display these observations. Overall, the PCA separated most samples in the Omicron/HC cohorts from the Wuhan/Delta cohorts in a two-dimensional plot (Fig. 4F). Further analysis using the PCA revealed that all infected individuals with the Wuhan strain were separated from HCs except for two patients (Fig. 4G). A similar pattern was observed when the plasma Gal levels were compared in HCs versus those infected with the Delta variant (Fig. 4H).

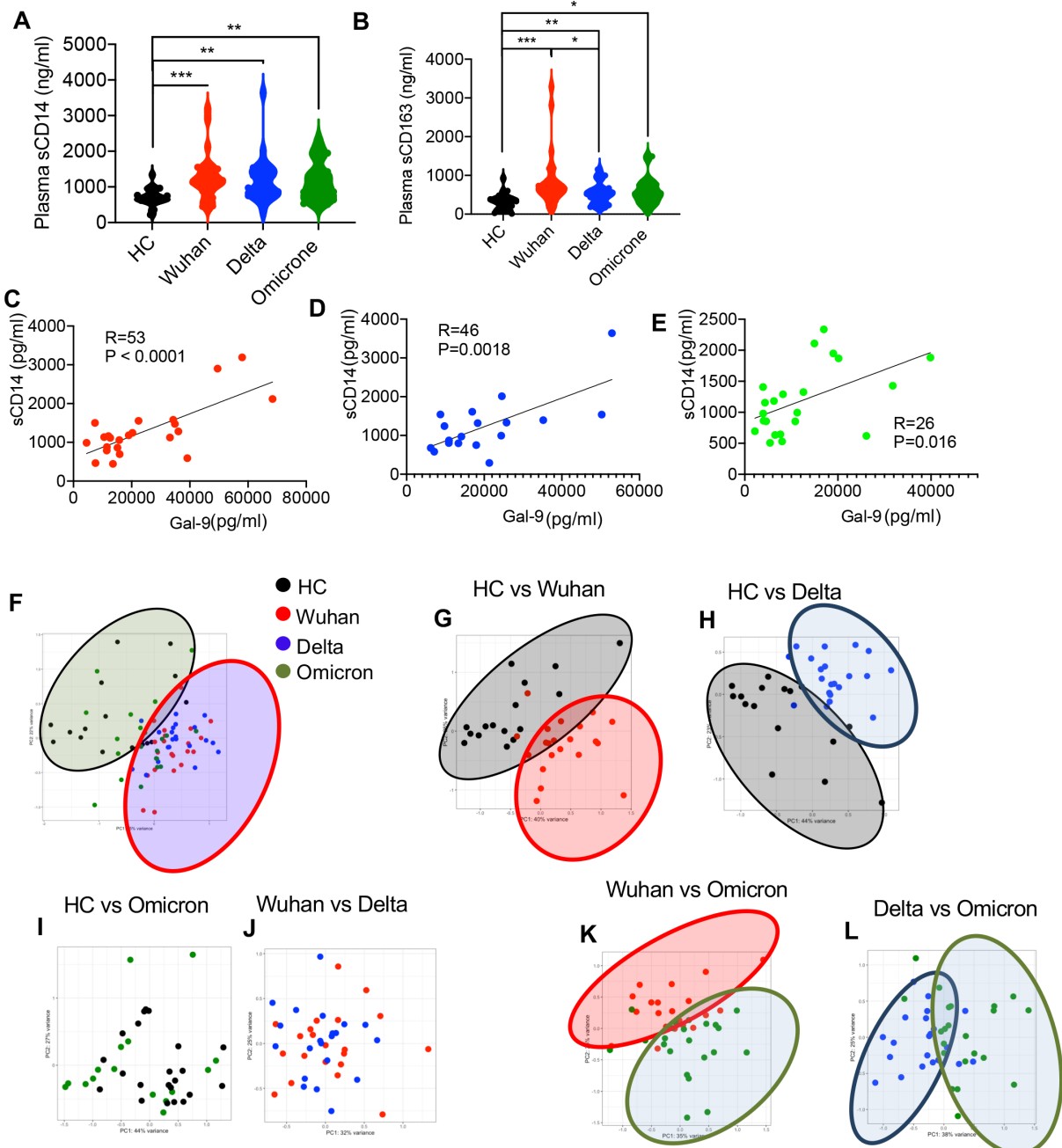

**FIG 4** The elevation of plasma sCD14 and sCD163 in COVID-19 patients infected with the Wuhan strain vs the Delta and Omicron variants. (A) Detected concentrations of sCD14, and (B) sCD163 in plasma samples of HCs vs individuals infected with the SARS-CoV-2 original strain or the Delta and Omicron variants as measured by ELISA. Correlation of sCD14 and Gal-9 concentrations in the plasma samples of (C) the Wuhan cohort, (D) the Delta cohort, and (E) the Omicron cohort. The PCA on the Euclidian distances between Gals, sCD14, and sCD163 of plasma samples of (F) HCs vs three COVID-19 cohorts, (G) HC vs Wuhan, (H) HC vs Delta, (I) HC vs Omicron, (J) Wuhan vs Delta, (K) Wuhan vs Omicron, and (L) Delta vs Omicron variants. *, $P < 0.05$; **, $P < 0.01$; ***, $P < 0.001$; ****, $P < 0.0001$.

However, we did not observe any distinct clustering between HCs and those infected with the Omicron variant in terms of plasma Gals and sCD14/sCD163 levels (Fig. 4I). This was also the case for infected individuals with the Wuhan strain versus the Delta variant (Fig. 4J). Finally, these analyses revealed that infected individuals with the Wuhan strain and the Delta variant were mostly separated from those infected with the Omicron in terms of soluble Gals and sCD14/sCD163 (Fig. 4K and L). In summary, the plasma profile for Gals, sCD14, and sCD163 of HCs was distinct from infected individuals with the Wuhan and Delta variants, but patients infected with the Omicron variant had a profile more similar to HCs. Considering the similarity between the plasma concentrations of sCD14 and sCD163 in infected individuals with different SARS-CoV-2 variants, the differences observed in PCAs could be largely driven by Gals. Hence, given the stimulatory functions of Gals on innate immune cells (e.g., NK cells, monocytes, and macrophages) (6, 37, 38), a lower plasma Gal in infected individuals with the Omicron variant may, in part, explain a milder innate immune response. In summary, these observations indicate differential effects of SARS-CoV-2 viral strains/variants on the abundance of soluble Gals and sCD14/sCD163 in the plasma of ICU-admitted COVID-19 patients.

## Differential abundance of CD4[+] T cells expressing immune checkpoints and activation markers in SARS-CoV-2-infected individuals with different viral strains/variants

We and others have reported that the frequency of T cells expressing different co-inhibitory and co-stimulatory receptors was expanded in individuals infected with the Wuhan strain (16, 18). Therefore, we compared the frequency of T cells expressing different immune checkpoints in three COVID-19 cohorts. Our results showed significant expansion of Gal-9 and PD-1 expressing CD4[+] T cells in PBMCs of all individuals infected either with the Wuhan strain or Delta/Omicron variants when compared with HCs (Fig. 5A through D). However, the frequency of TIGIT[+], TIM-3+, CD160+, and CD3[+]CD4[+] T cells was significantly higher in infected individuals with the Wuhan strain and the Omicron variant, but infected individuals with the Delta variant had a significantly lower frequency of these CD4[+] T cell subpopulations (Fig. 5E through L). In addition, our results showed that the frequency of 2B4+ (CD244) CD4[+] T cells was only enriched in patients infected with the Wuhan SARS-CoV-2 strain, not the other two variants (Fig. 5M and N). Conversely, CD73 expressing CD4[+] T cells were significantly lower in all three groups of COVID-19 patients compared to HCs, with patients infected with the Wuhan strain exhibiting a more pronounced depletion of CD73[+]CD4[+] T cells when compared with HCs and the other two COVID-19 cohorts (Fig. S2A and B).

Next, we investigated the activation status of CD4[+] T cells in our cohorts, which revealed a significant expansion of CD71[+], HLA-DR[+], CD38[+], and HLA-DR[+]CD38[+] CD4[+] T cells in patients infected with the Wuhan strain and Omicron variant compared with HCs (Fig. S2C and D). However, there was no significant difference in the frequency of CD71[+], HLA-DR[+], CD38[+], and HLA-DR[+]CD38[+]CD4[+] T cells in patients infected with the Delta variant when compared with HCs (Fig. S2C and D). Interestingly, infected individuals with the Wuhan strain and Omicron variant harbored significantly higher frequency of CD71[+], HLA-DR[+], CD38[+], and HLA-DR[+]CD38[+]CD4[+] T cells compared with patients infected with the Delta variant (Fig. S2C and D). Overall, our data indicate that CD4[+] T cells in patients infected with the Wuhan strain display greater activation phenotype compared with their counterparts in cohorts infected with the Omicron and Delta variants. It is worth mentioning that CD4[+] T cells in infected individuals with the Delta variant display the least activation.

## Differential abundance of CD8[+] T cells expressing immune checkpoints and activation markers in SARS-CoV-2-infected individuals with different viral strains/variants

Our results revealed a significant expansion of CD8[+] T cells expressing 2B4, TIM-3, CD39, Gal-9, PD-1, and TIGIT in patients infected with the Wuhan strain versus HCs (Fig. 6A

through L). However, this pattern was different in patients infected with the Delta variant. We found only a significant increase in the frequency of CD8[+] T cells expressing CD39 and Gal-9 without any change in the frequency of TIM-3, 2B4, PD-1, and TIGIT expressing CD8[+] T cells in Delta-infected individuals compared to HCs (Fig. 6A through L). Interestingly, those infected with the Omicron variant exhibited a greater frequency of CD8[+] T cells expressing TIM-3, CD39, Gal-9, PD-1, and TIGIT versus HCs (Fig. 6A through L). When the expression of co-inhibitory receptors and ligands between different patient cohorts was analyzed, we noted that those infected with the Delta variant had a significantly lower abundance of CD8[+] T cells expressing TIM-3, CD39, Gal-9, and PD-1 compared with those infected with the Wuhan strain while the frequency of 2B4+ cells remained unchanged (Fig. 6A through L). Surprisingly, we found that the expression pattern of co-inhibitory receptors in patients infected with the Omicron variant was more similar to those infected with the Wuhan strain than the Delta variant. As shown in Fig. 6A through L, the frequency of CD8[+] T cells expressing 2B4, TIM-3, PD-1, and TIGIT was reminiscent to infected individuals with the Wuhan strain and the Omicron variant. At the same time, we observed a significantly higher proportion of CD39 but a lower frequency of Gal-9 expressing CD8[+] T cells in Omicron-infected in comparison to Wuhan-infected individuals (Fig. 6A through L). Moreover, we found a significant reduction in the proportion of CD8[+] T cells expressing TIM-3, CD39, PD-1, and TIGIT without any changes in the frequency of 2B4 and Gal-9 expressing CD8[+] T cells in infected individuals with the Delta variant versus the Omicron variant (Fig. 6A through L). Finally, we quantified the frequency of CD160 and CD73 expressing CD8[+] T cells in our cohorts. These studies revealed that the frequency of CD160[+]CD8[+] T cells was significantly elevated in patients infected with the Omicron variant compared to those infected with either the original strain or the Delta variant (Fig. 6M and N). As anticipated according to another report (18), we found a significant reduction in the frequency of CD73[+]CD8[+] T cells in all infected individuals with SARS-CoV-2 regardless of the viral strain/variant (Fig. S2E and F).

Considering the differential pattern of co-inhibitory expressing CD8[+] T cells in different cohorts, we analyzed the frequency of activated T cells using CD71, CD38, and HLA-DR. We found a significant increase in percentages of CD8[+] T cells expressing CD71, HLA-DR, and HLA-DR/CD38 cells in patients infected with the Wuhan strain or the Omicron variant compared to HCs (Fig. S2G and H). In contrast, Delta-infected patients only showed a significant abundance in HLA-DR expressing CD8[+] T cells compared with HCs (Fig. S2G and H). Moreover, we noted a significant increase in the frequency of CD71[+]CD8[+] T cells in those infected with the original strain versus the other two variants (Fig. S2G and H). Notably, the percentages of CD38[+]/CD38[+]HLA-DR[+]CD8[+] T cells were significantly greater in those infected with the Omicron variant than those infected with either the original strain or the Delta variant (Fig. S2G and H).

Taken together, our observations indicate a differential pattern of co-inhibitory receptors/ligand expression and activation markers in patients infected with the original strain versus variants. In particular, we noted a less robust expansion of T cells expressing immune checkpoint molecules in infected individuals with the Delta variant compared to those infected with both the Wuhan strain and the Omicron variant.

## SARS-CoV-2 infection is associated with the elevation of sICs in the plasma

sICs are either cleaved from membrane-bound forms or via alternative splicing (21). However, they may maintain the functional properties of the membrane-bound forms upon interaction with their corresponding ligands/receptors and can modulate immune cell functions (15). Several studies have reported the abundance of a few sICs in the plasma of SARS-CoV-2-infected individuals (23) without segregating the effects of different variants. Thus, we quantified levels of different sICs including sTIM-3, sTIGIT, sPD-1, sOX40, sLAG-3, sCD40L, sCD28, sCTLA-4, sGITR, and sCD27 in the plasma of our cohorts. As we expected, the PCA on Euclidean distances between plasma samples of our four different cohorts separated HCs from SARS-CoV-2-infected individuals (Fig. 7A). More specifically, all patients infected with SARS-CoV-2 regardless of the viral strain/

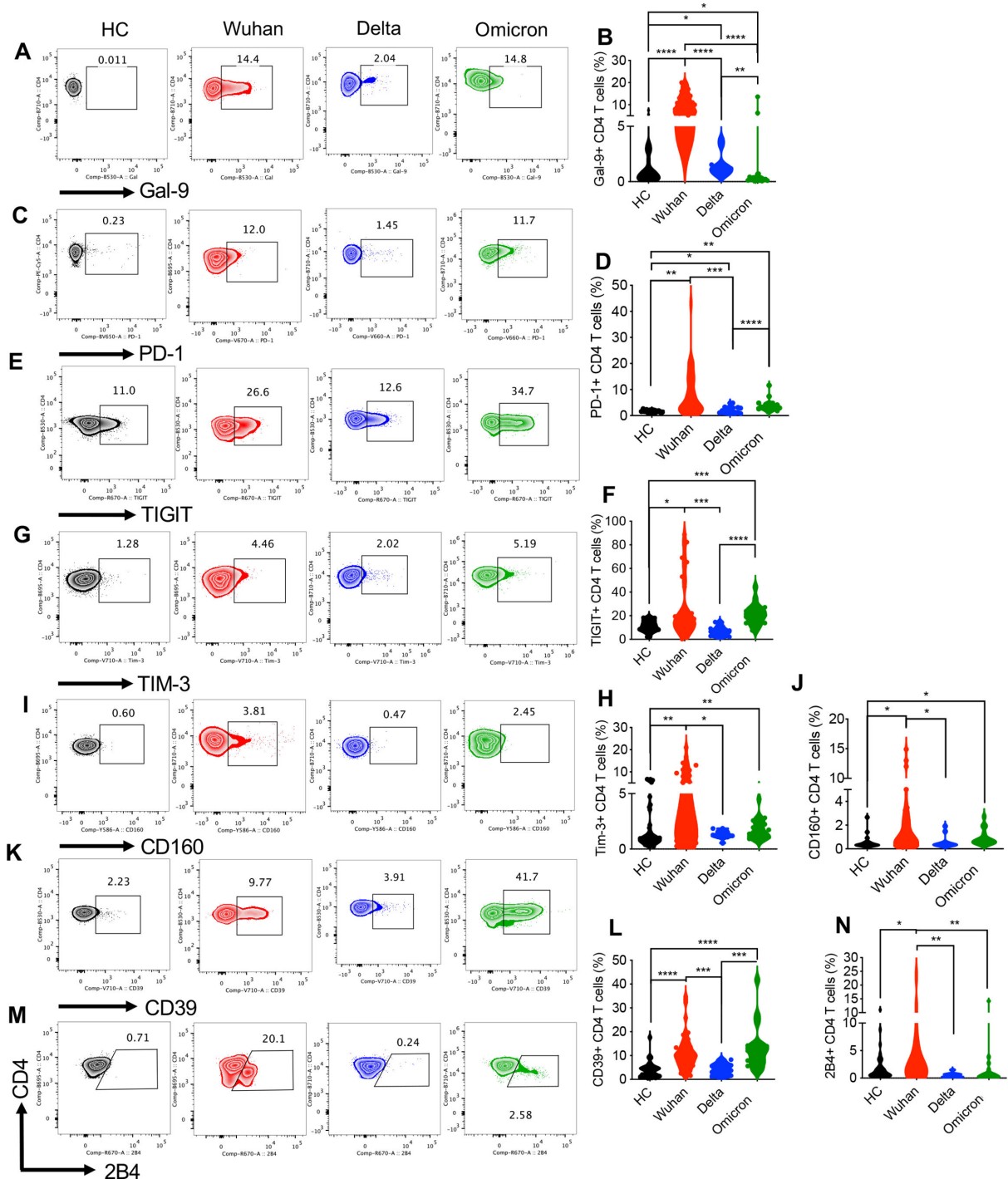

**FIG 5** Pronounced expansion of CD4[+] T cells expressing immune checkpoints in the Wuhan strain vs Delta and omicron variants. (A) Representative flow cytometry plots and (B) cumulative data of the percentages of Gal-9 expressing CD4[+] T cells in HCs vs individuals infected with the Wuhan strain, Delta variant, and Omicron variant. (C) Representative flow cytometry plots and (D) cumulative data of the percentages of PD-1 expressing CD4[+] T cells in HCs vs individuals infected with the Wuhan strain, Delta variant, and Omicron variant. (E) Representative flow cytometry plots and (F) cumulative data of the percentages of TIGIT expressing CD4[+] T cells in HCs vs individuals infected with the Wuhan strain, Delta variant, and Omicron variant. (G) Representative flow cytometry plots and (H) cumulative data of the percentages of TIM-3 expressing CD4[+] T cells in HCs vs individuals infected with the Wuhan strain, Delta variant, and Omicron variant. (I) Representative flow cytometry plots and (J) cumulative data of the percentages of CD160 expressing CD4[+] T cells in HCs vs individuals infected with the Wuhan strain, Delta variant, and Omicron variants (K) Representative flow cytometry plots and (L) cumulative data of the percentages of CD39 expressing CD4[+] T cells in HCs vs individuals infected with the Wuhan strain, Delta variant, and Omicron variants. (M) Representative flow cytometry plots and (N) cumulative data

**FIG 5** (Continued)

of the percentages of 2B4 (CD244) expressing CD4[+] T cells in HCs vs individuals infected with the Wuhan strain, Delta variant, and Omicron variant. *, $P < 0.05$; **, $P < 0.01$; ***, $P < 0.001$; ****, $P < 0.0001$.

variant had elevated levels of sTIM-3, sTIGIT, sPD-1, sLAG-3, and sCD27 in their plasma samples compared with HCs (Fig. 7B; Fig. S3A to C). However, only those infected with the Wuhan strain or the Omicron variant exhibited greater levels of sOX40 compared to uninfected individuals (Fig. 7B; Fig. S3A to C). For sCD40L, this pattern was different, and only infected individuals with the Delta variant showed elevated levels of this soluble ligand compared to HCs (Fig. 7B). The magnitude and differential levels of these soluble proteins in individuals infected with the Wuhan strain, the Delta variant, and the Omicron variant compared to HCs are illustrated in volcano plots, respectively (Fig. 7C through E). Such analysis revealed significantly increased levels of sCD40L and LAG-3 but decreased levels of OX40 and CD27 in the plasma of individuals infected with the Delta variant versus the Wuhan strain (Fig. 7F; Fig. S3D). When the levels of these sICs in individuals infected with the Delta and Omicron variants were assessed, we noted only a significant increase in OX40 but a decrease in CD40L (Fig. 7G; Fig. S3E).

To further present our observations in a more meaningful way, we performed the PCA. Altogether, patients infected with either the Wuhan strain or the Delta/Omicron variants were clearly distinct from HCs (Fig. 7H through J). However, when infected patients with different SARS-CoV-2 strains/variants were subjected to the PCA, we noted that there were a few overlaps between individuals infected with the Wuhan strain or Delta variant and similarly between the Delta or Omicron variant, respectively (Fig. 7K and L). Surprisingly, there was no clear separation between those infected with the Wuhan strain or the Omicron variant (Fig. 7M). Although infected individuals with SARS-CoV-2 had elevated levels of sICs and were separated from HCs, infection with the Delta variant was associated with a different profile of plasma sICs compared with the other two SARS-CoV-2 strains/variants. Of note, the levels of sCTLA-4 and sGITR were over the detection limit and, therefore, excluded.

## Pretreatment of Vero E6 cells with Gal-3, Gal-8, and Gal-9 enhances their susceptibility to infection with pseudo-SARS-CoV-2

We found that overnight treatment with Gal-3, Gal-8, or Gal-9 (2 µg/mL) increased the susceptibility of Vero E6 cells to pseudo-SARS-CoV-2 infection. This was illustrated in percentages and the absolute number of infected Vero E6 cells (Fig. 8A through C). Although we did not observe a dose-dependent effect for Gal-3 and Gal-8, Gal-9 enhanced the infection in a dose-dependent manner (Fig. 8A through C). A similar observation was made for the intensity of infection in these cells (Fig. S3F and G). Cells that were not exposed to pseudo-SARS-CoV-2 and Gals served as controls. Moreover, to better understand the potential mechanism(s) associated with the enhanced infection of Vero E6 cells with Gals, we further examined Gal-9, the post potent Gal (Fig. 8A through C). We found that although Gal-9 significantly increased psedu-SARS-CoV-2 infection, its pre-treatment of Vero E6 cells did not exert any additional effects once compared with simultaneous addition of Gal-9 (Fig. 8D through F). These observations suggest that Gal-9 likely does not upregulate the SARS-CoV-2 receptor/coreceptor but facilitates its entry via interaction with the virus, as reported for HIV (39). Vero E6 cells are permissible to SARS-CoV-2 infection and widely used for viral isolation and *in vitro* studies (40).

## DISCUSSION

SARS-CoV-2 variants exhibit different viral transmissibility and disease severity in infected individuals (4, 5). We have reported that the original strain of SARS-CoV-2 has a more profound effect on erythropoiesis compared to its more evolved counterparts, namely the Delta and Omicron variants (41, 42). Here, we provide additional insight into the differential effects of the Wuhan strain versus the Delta and Omicron variants on various

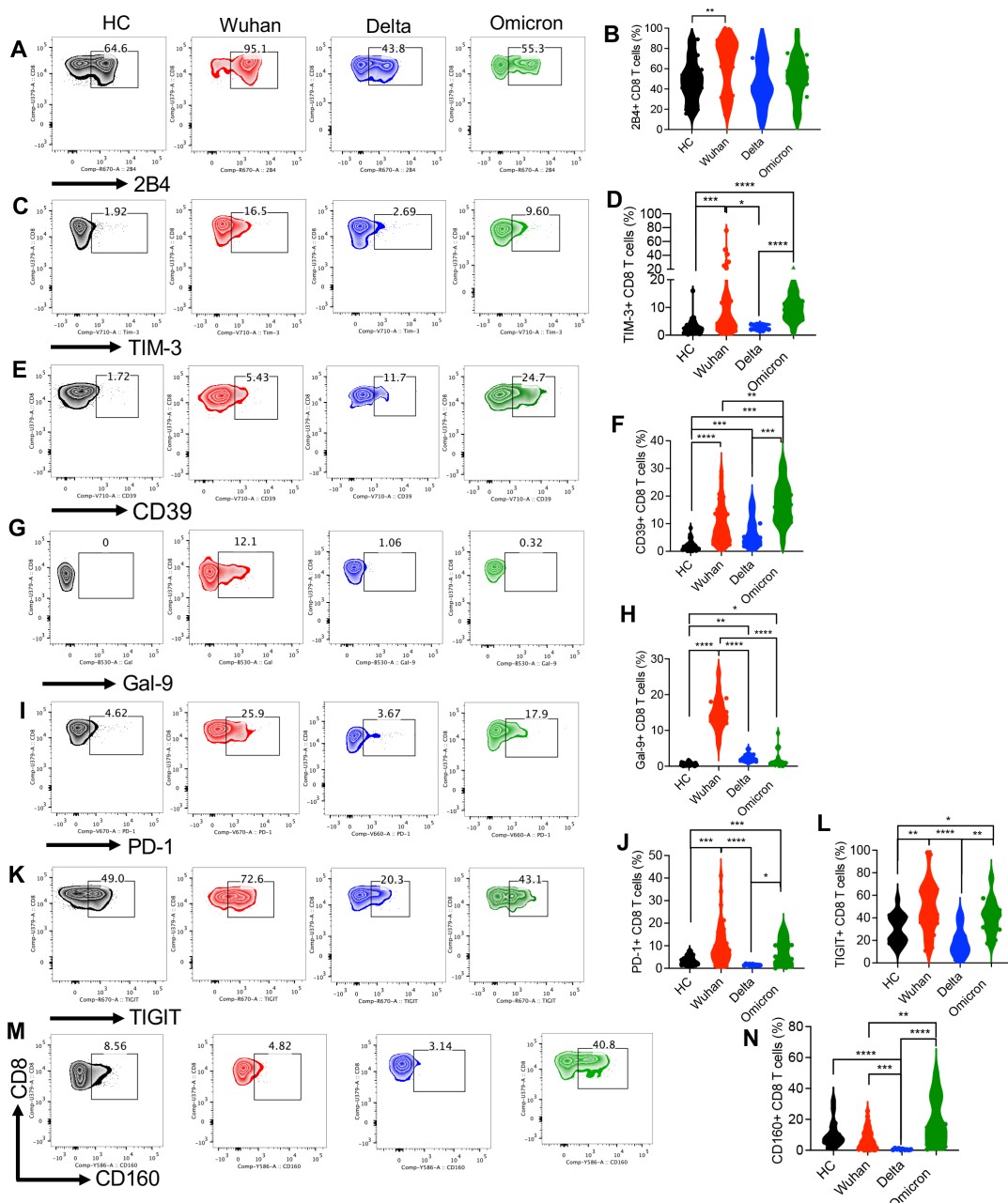

**FIG 6** Prominent expansion of CD8[+] T cells expressing immune checkpoints in the Wuhan strain vs Delta and Omicron variants. (A) Representative flow cytometry plots and (B) cumulative data of the percentages of 2B4 expressing CD8[+] T cells in HCs vs individuals infected with the Wuhan strain, Delta variant, and Omicron variant. (C) Representative flow cytometry plots and (D) cumulative data of the percentages of TIM-3 expressing CD8[+] T cells in HCs vs individuals infected with the Wuhan strain, Delta variant, and Omicron variant. (E) Representative flow cytometry plots and (F) cumulative data of the percentages of CD39 expressing CD8[+] T cells in HCs vs individuals infected with the Wuhan strain, Delta variant, and Omicron variant. (G) Representative flow cytometry plots and (H) cumulative data of the percentages of Gal-9 expressing CD8[+] T cells in HCs vs individuals infected with the Wuhan strain, Delta variant, and Omicron variant. (I) Representative flow cytometry plots and (J) cumulative data of the percentages of PD-1 expressing CD8[+] T cells in HCs vs individuals infected with the Wuhan strain, Delta variant, and Omicron variant. (K) Representative flow cytometry plots and (L) cumulative data of the percentages of TIGIT expressing CD8[+] T cells in HCs, individuals infected with the Wuhan strain, Delta variant, and Omicron variant. (M) Representative flow cytometry plots and (N) cumulative data of the percentages of CD160 expressing CD8[+] T cells in HCs vs individuals infected with the Wuhan strain, Delta variant, and Omicron variant. *, $P < 0.05$; **, $P < 0.01$; ***, $P < 0.001$; ****, $P < 0.0001$.

components of the immune system in ICU-admitted COVID-19 patients. We found that the Wuhan strain and the Delta variant promote a similar pattern of cytokine and

chemokine production when compared to uninfected HCs. However, the Omicron variant was associated with a milder and less pronounced cytokine and chemokine production compared with HCs. More specifically, our observations demonstrate that individuals infected with the Omicron variant had significantly lower levels of plasma IL-18, IP-10, IL-7, VEGF-A, MIP-1α, MCP-1, IL-8, Eotaxin, TNF-α, IFN-γ, and IL-1β compared

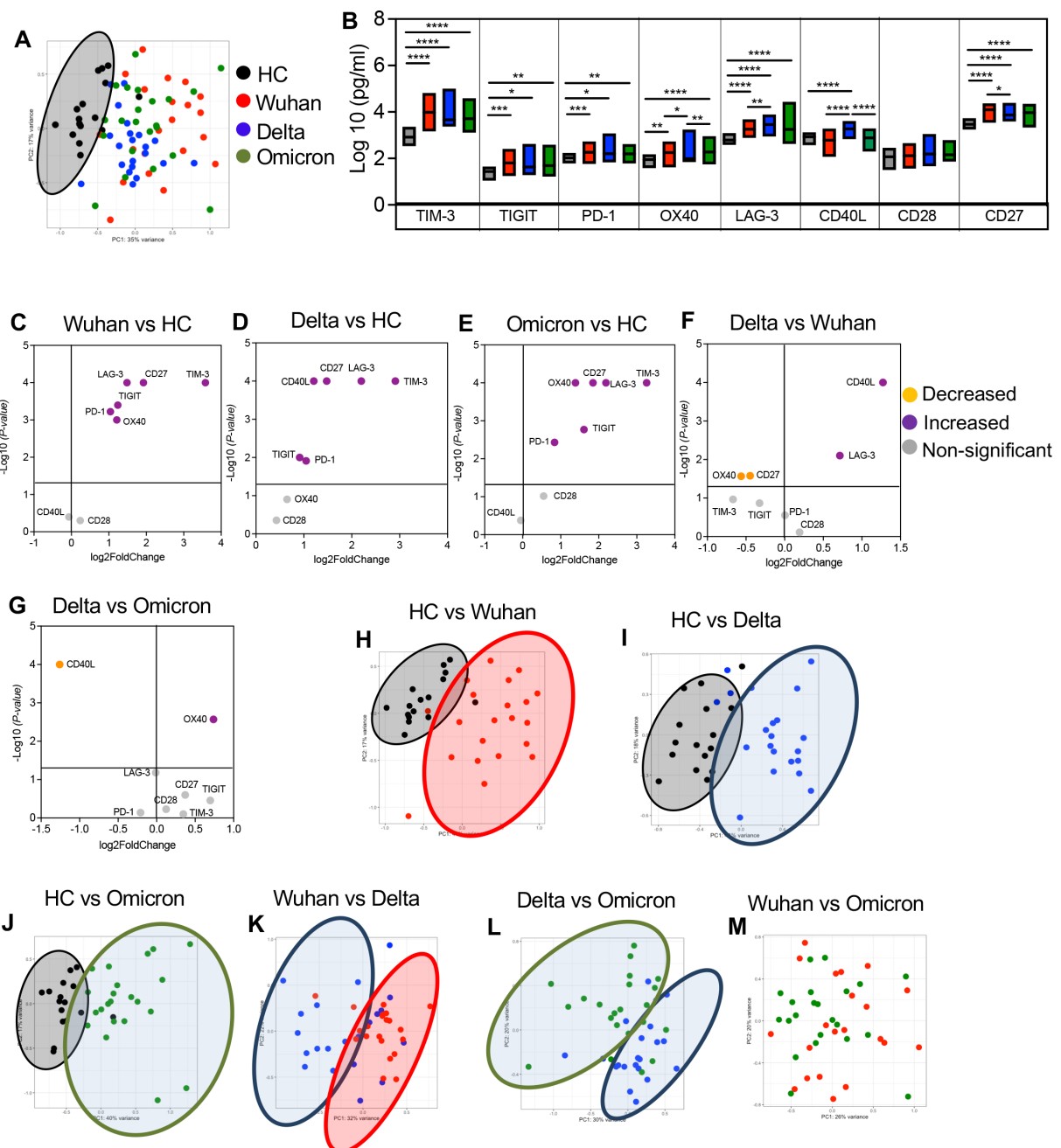

**FIG 7** The elevation of soluble immune checkpoints (sICs) in the plasma of individuals infected with SARS-CoV-2. (A) The PCA on the Euclidian distances between sICs in the plasma of our different study cohorts. (B) Normalized and calculated concentrations of sICs in the plasma of COVID-19 patients infected with different strains/variants of SARS-CoV-2 (n = 62) vs HCs (n = 16) as measured by MSD. Volcano plot illustrating the magnitude and significance of the differences in sICs concentrations in (C) Wuhan vs HC, (D) Delta vs HC, (E) Omicron vs HC, (F) Delta vs Wuhan, and (G) Delta vs Omicron variant. The significant difference in median plasma concentration was calculated for multiple testing. The PCA on the Euclidian distances between sICs of plasma samples of (H) HC vs Wuhan, (I) HC vs Delta, (J) HC vs Omicron, (K) Wuhan vs Delta, (L) Delta vs Omicron, and (M) Wuhan vs Omicron variant. *, P < 0.05; **, P < 0.01; ***, P < 0.001; ****, P < 0.0001. Circles marked in purple and orange are significantly increased or decreased, respectively. The circles in gray indicate no significant difference.

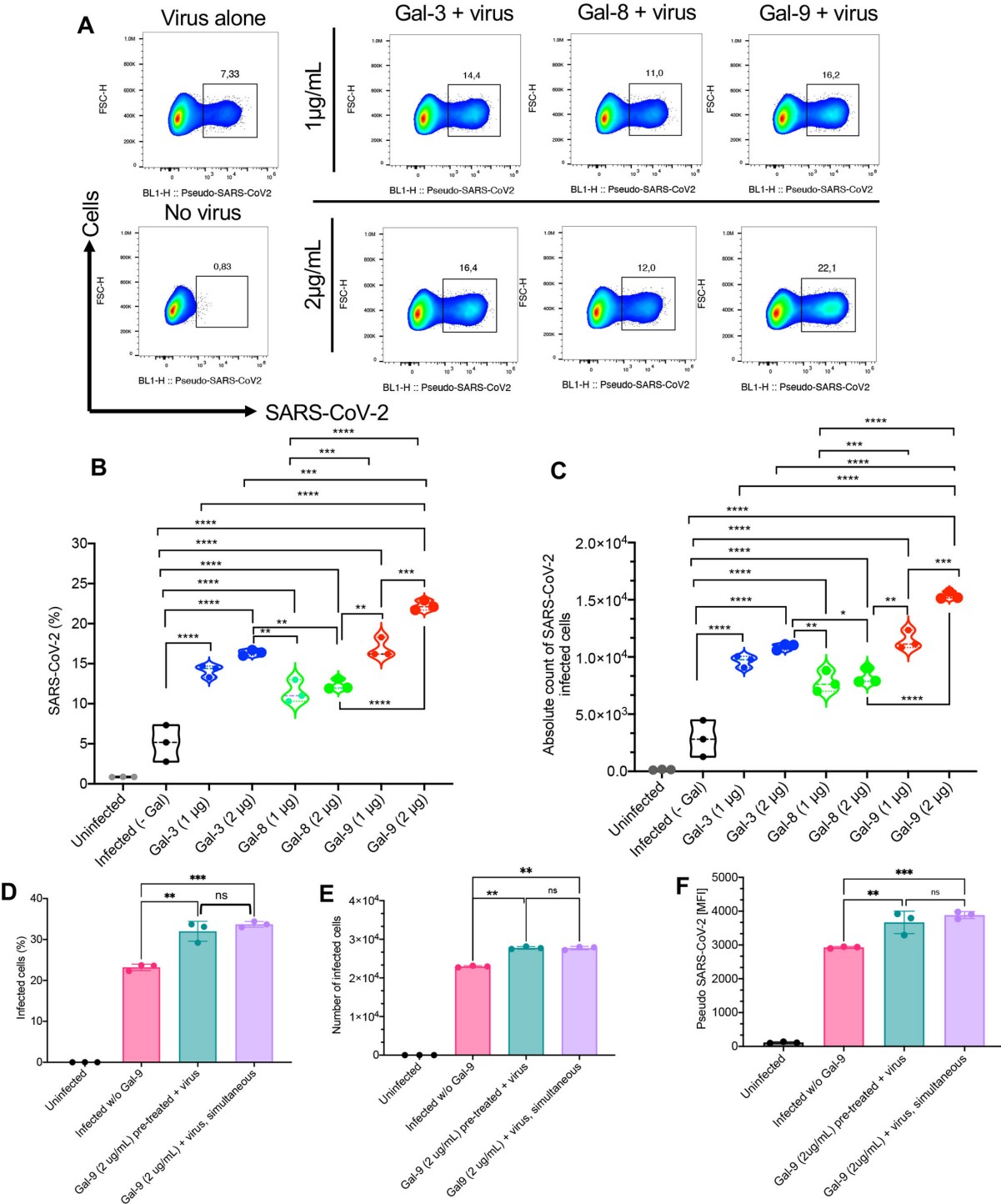

**FIG 8** Galectins enhance SARS-CoV-2 infection. (A) Representative flow cytometry plot and (B) cumulative data of percentages of SARS-CoV-2-infected cells in the absence or presence of pseudovirus with or without hrGal-3, hrGal-8, and hrGal-9 at indicated concentrations. (C) Cumulative data showing the absolute number of infected cells with pseudo-SARS-CoV-2 in the absence or presence of hrGal-3, hrGal-8, and hrGal-9 at indicated concentrations. (D) Cumulative results of the percentages of infected cells, (E) absolute number of infected cells, and (F) the intensity of infection (mean fluorescence intensity) with pseudo-SARS-CoV-2 in the presence of Gal-9 (2 µg/mL) either pre-treated overnight or added to the cell with the pseudo virus simultaneously and analyzed after 18 h. **, $P < 0.01$; ***, $P < 0.001$. ns, no significant.

to those infected with the Wuhan strain. Similar analysis revealed lower levels of cytokines and chemokines (e.g., Eotaxin-3, IP-10, IL-7, Eotaxin, VEGF-A, MCP-1, IFN-γ, and

IL-1β) in individuals infected with the Omicron or the Delta variant. These results suggest that the Omicron variant is linked to a milder cytokine release syndrome compared to the Wuhan strain and the Delta variant. The attenuated cytokine storm may explain less severe disease and lower mortality in individuals infected with the Omicron variant in comparison to the original strain and the Delta variant (4, 5). Therefore, structural changes due to mutations in the spike protein may result in lower immune activation (43) and subsequently a less pronounced cytokine release syndrome in individuals infected with the Omicron variant. This is in line with reports of attenuated disease in animal models of SARS-CoV-2 infection with the Omicron variant (44, 45) and reduced odds of severe disease compared with individuals infected with the Delta variant (46). In contrast, a recent study has claimed that the Omicron variant is equally efficient in promoting cytokine production compared to the Wuhan strain and Delta variant (47). Nevertheless, this study did only test the N-terminus domain of the S1 subunit of the Omicron on isolated PBMCs *in vitro*. To summarize, this study does not mimic the physiological circumstance, as PBMCs are not the only source of cytokines and chemokines and other immune (e.g., polymorphonuclear cells) and non-immune cells but also contribute to the cytokine storm in COVID-19 patients. As such, we propose that a lower cytokine storm in individuals infected with the Omicron may prevent the alveolar damage and respiratory failure commonly observed in individuals infected with the original strain or the Delta variant of SARS-CoV-2 (48).

It should be noted that it is challenging to distinguish an appropriate hyperimmune response against the infection from a dysregulated immune response associated with collateral damage. To add another layer of complexity, most cytokines/chemokines exhibit pleotropic downstream effects in a clinical setting and are frequently synergistic in their biological properties. This complex interplay demonstrates that cytokines/chemokines and other mediators are neither linear nor consistent. It is worth mentioning that although their elevated levels imply severity of response to a specific SARS-CoV-2 variant, they do not automatically support pathogenesis. Moreover, SARS-CoV-2-infected cells (e.g., monocytes) can activate inflammasome and trigger pyroptosis (49) in which infected cell loses cell membrane integrity and releases a wide range of cytokines and chemokines including cytosolic Gals. In line with this hypothesis, we noted elevated levels of Gal-3, Gal-8, and Gal-9 concentrations in the plasma of individuals infected with the Wuhan strain followed by the Delta and Omicron variants of SARS-CoV-2. Gals are widely expressed intracellularly or extracellularly in a wide range of immune and non-immune cells (e.g., lung epithelial cells and endothelial cells) (28). They might source from the virally infected cells or cell-associated damage/lysis resulting in their leakage from the cytoplasm. Gals such as Gal-3, Gal-8, and Gal-9 upon interaction with different receptors in an autocrine and/or paracrine manner promote the induction of a wide range of proinflammatory cytokines and chemokines (6, 50, 51). For example, Gal-3 is reported to be associated with several inflammatory and thrombo-inflammatory analytes in COVID-19 patients (50), and its inhibition has been considered a therapeutic approach in hospitalized COVID-19 patients (35). Similarly, Gal-9 has been reported as a surrogate biomarker associated with the cytokine storm in those infected with the Wuhan strain (6, 52). In addition to the release of cytosolic Gal-9 from damaged immune and non-immune cells, activated neutrophils shed Gal-9 (53, 54), which, upon interaction with CD44 or immune synapses, acts as a regulator of NK and T cell activation (38, 53, 55). The translocation of intracellular Gal-9 into immune synapses in activated T cells has been documented in HIV and cancer (32, 56). Likewise, Gal-8 in the same manner upon binding to CD44 triggers a cascade of signaling pathways leading to the induction of proinflammatory cytokines in COVID-19 patients (51). These observations support the notion that the elevated levels of plasma Gals may contribute to the cytokine release syndrome in COVID-19 patients. As such, it is likely to hypothesize that the lower levels of plasma Gals attenuate the cytokine release syndrome and minimizing immunopathology in individuals infected with the Omicron variant. On contrary, some may argue that a milder immune activation and subsequently a reduced infection associated direct or

indirect cell damage are the potential mechanism associated with lower Gal levels in individuals infected with the Omicron variant.

In support of the former hypothesis, our further analysis showed a strong correlation between the plasma Gals levels with prominent proinflammatory cytokines (e.g., IL-6, TNF-α, MCP-1, IP-10, MIP-1α, MIP-1β, Eotaxin, IL-8, GM-CSF, and MDC) in individuals infected with the Wuhan strain as reported elsewhere (6, 57). However, a few cytokines and chemokines (e.g., IL-18, MIP-1β, IL-6, TNF-α, and MCP-1) displayed correlation with Gal levels in Delta-infected individuals. Surprisingly, the association of Gal-9 levels with IL-18 was only observed in those infected with the Omicron variant. These observations imply that the plasma Gals in COVID-19 patients may serve as damage-associated molecular patterns (58) that exert their actions on diverse immune cells (e.g., mono-cytes/macrophages, NK cells, T cells, and neutrophils) to intensify the cytokine storm (6). In agreement, we found enhanced induction of IL-6 and TNF-α by rhGal-3, rhGal-8, and rhGal-9 in PBMCs, in particular, monocytes, although it was more prominent by rhGal-3, as reported elsewhere (35, 50). In addition, we found increased levels of sCD14 and sCD163 in the plasma of COVID-19 patients, which support an essential role for activated monocytes in SARS-CoV-2 pathogenesis (24). This concept was further promoted by a positive correlation between Gal-9 levels with sCD14 in COVID-19 patients, which was more pronounced in the Wuhan strain cohort. Overall, our results indicate differ-ential effects of Gals in individuals infected with SARS-CoV-2 strains/variants, which does not necessarily explain their plasma levels. Alternatively, SARS-CoV-2 variants may differentially modulate proteolysis of Gals and subsequently influence the host defense mechanism. For instance, Gal-8 cleavage by SARS-CoV-2 VoC has been considered an immune response escape mechanism (59).

Another possibility is that Gals may directly interact with SARS-CoV-2 and subse-quently influence its infectivity as illustrated for Gal-3 and Gal-9 in HIV infection (39, 60). For example, the plasma levels of Gal-9 are correlated with the viral load in acute dengue virus (28) and HIV infections (30). In contrast, Gal-8 acts as an intracellular sensor for SARS-CoV-2 in promoting protective xenophagy (26). In agreement, our results support a role for Gals in SARS-CoV-2 infection. We found that in all studies on Gals, Gal-9, in particular, enhances SARS-CoV-2 infection possibly through enhanced viral entry as reported in HIV (39). Further studies on Gal levels and viral load in the endotracheal aspirates may shed light on the role of Gals in viral replication.

Broadly speaking, Gals denote a hyperactive immune response characterized by the elevation of cytokines, chemokines, and several other mediators as part of the innate immune response against SARS-CoV-2 infection (61). Subsequently, this inflammatory milieu may promote T cell activation in an antigen-independent or/and T cell receptor (TCR)-independent manner (62). In addition, the antigen-dependent mechanism may contribute to T cell activation in individuals infected with SARS-CoV-2. Consistent with this hypothesis, we observed the expansion of activated T cells expressing CD38, CD71, and HLA-DR, which was more prominent in those infected with the Wuhan strain.

In agreement with the notion of enhanced T cell activation, we observed an expansion of T cells expressing co-inhibitory/stimulatory receptors. However, this was more distinct/pronounced in those infected with the Wuhan strain compared to those infected with either the Delta or Omicron variant. Notably, increased frequency of T cells expressing co-inhibitory receptors does not necessarily support an exhausted T cell phenotype (63), as T cell exhaustion is characterized by functional impairment of T cells (15). Previously, we reported that T cells expressing co-inhibitory receptors (e.g., PD-1, TIM-3, CD160, Gal-9, etc.) did not exhibit an impaired phenotype in SARS-CoV-2-infected individuals (18). Therefore, we believe that transient expression of co-inhibitory receptors on activated T cells in SARS-CoV-2-infected individuals is meant to prevent an excessive hyperimmune response. For instance, the PD-1/PDL-1 axis can mediate potent inhibitory signals to reduce T effector cell function as a protective mechanism for preventing collateral damage, without compromising antiviral immunity in an acute setting (64). In

fact, the massive release of inflammatory cytokines in T cells expressing co-inhibitory receptors (18) is very similar to what occurs during T cell activation (65).

In contrast to the expansion of CD39$^+$ T cells, we noted a significant reduction in the frequency of CD7$^+$ T cells in SARS-CoV-2-infected individuals as reported in HIV infection (66). The purinergic system via CD39 and CD73 ecto-enzymes fine-tunes immune cell activation. This system shifts an ATP-driven proinflammatory to an anti-inflammatory milieu mediated by adenosine (67). This dysregulated expression level of these two ecto-enzymes in COVID-19 patients may promote an inflammatory response that influences lymphocyte homing to draining lymph nodes (68). Surprisingly, we noted moderate T cell activation in individuals infected with the Delta variant despite comparable levels of proinflammatory cytokines/chemokines with the Wuhan strain.

Although elevated levels of cytokines and viral load may delay SARS-CoV-2-specific T cell responses in infected individuals with the Delta variant (69), in our studies, we analyzed total T cells regardless of their antigen specificity. Notably, we are unaware of the viral load in different patient cohorts to determine whether viral load influences T cell response. It is fair to speculate that lower T cell activation could be an escape mechanism deployed by the Delta variant to evade the adaptive immunity. Although T cell responses to the Omicron variant compared with the original strain are reported to be reduced, they are not different between the Delta and Omicron variants (70). However, this study reported antigen-specific T cell but not bulk T cells that were analyzed in our study.

In support of these observations, we found that the pattern of sICs in individuals infected with the Delta variant was different from those infected with the ancestral strain and the Omicron variant. These circulating factors such as sICs may represent immune parameters to evaluate the dynamic behavior of T cells in different patients' subpopulations. Notably, we detected significantly lower plasma levels of OX40 and CD27 in those infected with the Delta variant. Although the cell surface was not examined for these costimulatory receptors, our observations imply that reduced soluble forms of these two co-stimulatory molecules may explain lower T cell activation in the Delta cohort. Because CD27 and OX40 can support the survival of activated T cells (71), their declining levels may explain reduced T cell activation in the Delta-infected individuals. Dissecting the role of T cell-mediated immunity following infection is a significant mechanistic matter for the vaccine's efficacy. It is also valuable to know how T cell response impacts SARS-CoV-2 acquisition and disease severity. Although current knowledge does not carry an important role for T cell response against SARS-CoV-2 acquisition, T cells may prevent severe clinical outcomes in infected individuals (72). This concept has been described in influenza virus infection, where T cells are shown to prevent disease severity without influencing virus acquisition (73). Nevertheless, T cells are involved in supporting antibody responses, and likewise, B cells can modulate T cell effector functions (74). It also appears that differences in viral factors play a role in the differential immune responses in SARS-CoV-2-infected individuals as illustrated for erythropoiesis (41). Therefore, understanding which immune responses can mediate protection and/or prevent immunopathology is a top priority. Our observations suggest that a dysregulated innate immune response in individuals infected with the Wuhan strain or the Delta variant is the driving force for severe COVID-19 disease. Subsequently, Gals may enhance disease severity through intensifying the release of proinflammatory cytokines and chemokines. They can also directly interact with the SARS-CoV-2 viruses and affect viral replication. Alternatively, Gals may modulate antigen presentation/processing and stimulate the elicitation of T cell response in a lectin-dependent manner (75). Further studies are required to better understand the role of viral factors and Gals in the induction of cytokine release syndrome and provide evidence to better understand SARS-CoV-2 pathogenesis. We are aware of multiple study limitations such as inability to measure T cell effector functions due to the limited sample size. Analyzing the co-expression of co-inhibitory receptors on total T cells is very valuable, but we were unable to run all the co-inhibitory receptors on a single panel. In addition, due to the limited cell number, we were unable to analyze the expression of co-inhibitory/stimulatory receptors

on antigen-specific T cells. Although our *in vitro* studies suggest that Gals enhance viral infection, we did not have patients' nasal swab viral load or cycle threshold (Ct) values to correlate with the plasma Gals and cytokines. Another study limitation was our focus on hospitalized patients; therefore, it is impossible to determine whether such a differential immune response (e.g., cytokines, Gals, co-inhibitory receptor expression, etc.) can be observed in those infected with different viral strains/variants but presenting a milder disease.

## ACKNOWLEDGMENTS

We thank all the volunteers who supported this study by donating their samples and dedicating their time. We would also like to thank the flow cytometry core facility of the Faculty of Medicine and Dentistry at the University of Alberta.

S.S. performed most of the studies, analyzed the data, designed the figures, and wrote the Methods and Results sections. N.B. conducted part of the study and analyzed related data. J.L. performed Gals-related pseudovirus infection studies and analyzed related results. M.O. and W.S. clinician scientists recruited COVID-19 patients and provided clinical data. D.L.T. provided resources and advice on the study. S.E. conceptualized and designed the study, secured resources, assisted in figure design and data analysis, and wrote the manuscript.

This work was supported by the Canadian Institute for Health Research (CIHR#453061) through a project grant (to S.E.). Nevertheless, the funding bodies had no role in the design of the study, data collection, analysis, and publication.

The author has no relevant financial or non-financial interests to disclose.

## AUTHOR AFFILIATIONS

[1]Division of Foundational Sciences, School of Dentistry, University of Alberta, Edmonton, Alberta, Canada

[2]Division of Rheumatology, Department of Medicine, University of Alberta, Edmonton, Alberta, Canada

[3]Department of Critical Care Medicine, University of Alberta, Edmonton, Alberta, Canada

[4]Division of Infectious Diseases, Department of Medicine, University of Alberta, Edmonton, Alberta, Canada

[5]Department of Medical Microbiology and Immunology, University of Alberta, Edmonton, Alberta, Canada

[6]Li Ka Shing Institute of Virology, University of Alberta, Edmonton, Alberta, Canada

[7]Women and Children Health Research Institute (WCHRI), University of Alberta, Edmonton, Alberta, Canada

[8]Glycomics Institute of Alberta, University of Alberta, Edmonton, Alberta, Canada

[9]Alberta Transplant Institute, University of Alberta, Edmonton, Alberta, Canada

## AUTHOR ORCIDs

Shokrollah Elahi  http://orcid.org/0000-0002-7215-2009

## FUNDING

| Funder | Grant(s) | Author(s) |
| --- | --- | --- |
| Gouvernement du Canada \| Canadian Institutes of Health Research (IRSC) | 453061 | Shokrollah Elahi |

## AUTHOR CONTRIBUTIONS

Shima Shahbaz, Data curation, Formal analysis, Investigation, Methodology, Software, Visualization, Writing – original draft | Najmeh Bozorgmehr, Data curation, Formal analysis | Julia Lu, Data curation, Formal analysis | Mohammed Osman, Resources, Writing – review and editing | Wendy Sligl, Resources, Writing – review and editing | D. Lorne

Tyrrell, Resources, Writing – review and editing | Shokrollah Elahi, Conceptualization, Funding acquisition, Project administration, Resources, Supervision, Writing – original draft, Writing – review and editing

## DATA AVAILABILITY

All the relevant data are included in the main manuscript and supplemental materials.

## ETHICS APPROVAL

The Human Research Ethics Board (HREB) at the University of Alberta approved this study with protocol #Pro00099502. In addition, the HREB approved blood collection from HCs (protocol #Pro00063463). The study participants (HCs) provided written consent, and ICU patients provided verbal consent to be part of the study.

## ADDITIONAL FILES

The following material is available online.

### Supplemental Material

**Supplemental Figures (Spectrum01256-23-s0001.pdf).** Supplemental Figures 1-3
**Supplemental Table 1 (Spectrum01256-23-s0002.pdf).** S Table 1

### Open Peer Review

**PEER REVIEW HISTORY (review-history.pdf).** An accounting of the reviewer comments and feedback.

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
