## [Reviewer comments · Microbiology Spectrum]

Microbiology Spectrum

Analysis of SARS-CoV-2 isolates namely the Wuhan strain, Delta and Omicron variants identifies differential immune profiles

Shima Shahbaz, Najmeh Bozorgmehr, Julia Lu, Mohammad Osman, Wendy Sligl, D. Lorne Tyrrell, and Shokrollah Elahi

Corresponding Author(s): Shokrollah Elahi, University of Alberta

Review Timeline:

Submission Date:	March 22, 2023
Editorial Decision:	July 9, 2023
Revision Received:	July 13, 2023
Accepted:	July 13, 2023

Editor: Leonidas Stamatatos

Reviewer(s): The reviewers have opted to remain anonymous.

Transaction Report:

DOI: <https://doi.org/10.1128/spectrum.01256-23>

July 9, 2023

Dr. Shokrollah Elahi
University of Alberta
Dentistry and Medical Microbiology and Immunology
11361 87th Ave NW
Edmonton, AB T6G2E1
Canada

Re: Spectrum01256-23 (Analysis of SARS-CoV-2 isolates namely the Wuhan strain, Delta and Omicron variants identifies differential immune profiles)

Dear Dr. Shokrollah Elahi:

Dear Dr. Shokrollah Elahi,
Your manuscript was reviewed by two experts who have requested revisions to be made. Please address those comments and re-run your revised manuscript.

Best regards,
L. Stamatatos, Ph.D.

Link Not Available

Sincerely,

Leonidas Stamatatos

Journals Department
Reviewer comments:

Reviewer #1 (Comments for the Author):

In this study, Shabaz et al perform a multi-parameter analysis of cellular cytokines, chemokines and galectins on hospitalized ICU patients infected with Wuhan-Hu-1, Delta, and Omicron variants of SARS CoV2. All participants had higher levels of

cytokines, chemokines and galectins compared to healthy controls. However some notably lower cytokine levels were observed in individuals infected with Omicron as compared to Wuhan and Delta. Similarly some levels of galectins were lower in omicron as well. Overall the data suggest that omicron infection led to a lower activation of innate immune responses and a corresponding weaker cytokine storm. Interestingly the authors noted that in vitro infection of Vero cells with SARS Cov-2 pseudovirus was enhanced by galectin treatment.

Specific comments:

Was there any information available about whether the participants had prior SARS-CoV-2 infection? Since omicron emerged much later than Wuhan-1 and later than delta, its possible that in some donors, the lower levels of immune activation seen in omicron infection be due to pre-existing immune responses that better control infection.

Were the viral loads measured in the nasal swabs used to diagnose infection? Or were there differences in the Ct values? Does this correlate at all with cytokine or galectin levels?

One caveat with the study is that all the donors are hospitalized in the ICU so it's hard to determine whether lower levels of any of the parameters measured in omicron vs wuhan or delta translate to less severe disease.

Wuhan is misspelled repeatedly in Figures 2 and 4.

Reviewer #2 (Comments for the Author):

This study has a lot of interesting observations in it that are valuable due to overall good study design. I really appreciated that the authors timed the collections from patients to a standardized 2 weeks post start of symptoms or diagnosis of COVID, as that timing was quite appropriate for observations on the T cell compartment in particular. Also, having synced timing as well as having all patients on study being unvaccinated were critical for the observations as a whole.

It is not clear in the manuscript how the strain of COVID each patient on study had was determined - it should be stipulated whether the patient samples were subjected to strain-specific PCR tests, or if it was inferred based on timing of diagnosis relative to dominant strains at the time.

In Fig 1A, the PCA plot does not convincingly show a difference in the distance between HC and Wuhan strains versus patients with other strains as the authors describe in the text. There is a lot of spread in this data, so if that overall distance was determined mathematically and shows a significant difference, it should be shown, but otherwise I do not think that conclusion is valid. Figures 1E-G do support that conclusion by demonstrating clear differences in the cytokine levels in patient blood in different groups, but the PCA plot does not add to/support that conclusion, so I would recommend removing Fig 1A, or at least removing the conclusion that there is a clear difference in COVID infected groups from each other in that analysis.

In Figure 3, the figure legend refers to Fig 3I as showing IL6 levels, but the graph itself has TNFalpha instead on the axis. The opposite is the case for 3J.

There are a few concerning details in Figure 4 that need to be addressed. The PCA analysis done here includes as inputs both the galectin levels in the blood and the sCD14 and sCD163 data. It is not clear to me why comparisons including those particular data would be appropriate here, since Galectins are widely expressed. Galectin 3 is mainly derived from monocytes and mo-DCs, so that could be argued as related to the sCD14 measurements, but Gal 8 and Gal9 are widely expressed in non-immune cells.

Furthermore, in the figures for the PCA analysis, in Fig 4 F, I believe that "ICU" was mistakenly substituted for "Wuhan", which is confusing. Also, in the text of the manuscript, the authors indicate that the data suggests that infection with the Omicron variant is associated with "milder" activation of innate immune cells based on this PCA analysis, but this is not supported by the sCD14 data in Figure 4A, and in Fig 4B the sCD163 is lower than the Wuhan strain in both the Omicron patients and the Delta infected patients. Therefore, the differences seen in the PCA analysis seem to be driven largely by the inclusion of the Galectin data in that analysis, which is already detailed in Figure 2, and cannot be directly and solely tied to monocyte/macrophage activation.

This is a smaller point as I do think the figures and patient information table are clear on this point, but in the text there are a number of places where, when data is similar between different groups, phrases like "patients infected with Delta and Wuhan strains" are used. I find this a bit unclear as it sounds like the patients are infected with both strains at once, rather than referring to the two groups having similar data readouts - I would recommend rephrasing to "patients infected with Delta or Wuhan strains" (for example) in these cases.

Staff Comments:

Preparing Revision Guidelines

Please return the manuscript within 60 days; if you cannot complete the modification within this time period, please contact me. If you do not wish to modify the manuscript and prefer to submit it to another journal, please notify me of your decision immediately so that the manuscript may be formally withdrawn from consideration by Microbiology Spectrum.

Reviewer comments:

Reviewer #1 (Comments for the Author):

In this study, Shabaz et al perform a multi-parameter analysis of cellular cytokines, chemokines and galectins on hospitalized ICU patients infected with Wuhan-Hu-1, Delta, and Omicron variants of SARS CoV2. All participants had higher levels of cytokines, chemokines and galectins compared to healthy controls. However some notably lower cytokine levels were observed in individuals infected with Omicron as compared to Wuhan and Delta. Similarly some levels of galectins were lower in omicron as well. Overall the data suggest that omicron infection led to a lower activation of innate immune responses and a corresponding weaker cytokine storm. Interestingly the authors noted that in vitro infection of Vero cells with SARS Cov-2 pseudovirus was enhanced by galectin treatment.

We appreciate the supportive remarks about our work.

Specific comments:

Was there any information available about whether the participants had prior SARS-CoV-2 infection? Since omicron emerged much later than Wuhan-1 and later than delta, its possible that in some donors, the lower levels of immune activation seen in omicron infection be due to pre-existing immune responses that better control infection.

Thank you for the very important point. However, our patients did not have any previous evidence of symptomatic infection with earlier viral strain/variant. This has been included in page 6 lines 9-10 for clarification.

Were the viral loads measured in the nasal swabs used to diagnose infection? Or were there differences in the Ct values? Does this correlate at all with cytokine or galectin levels?

These are very important points, unfortunately, viral load in the nasal swab was not quantified and similarly Ct values were not available for all patients. This point is included in discussion as a study limitation (page 28 line 1-3).

One caveat with the study is that all the donors are hospitalized in the ICU so it's hard to determine whether lower levels of any of the parameters measured in omicron vs wuhan or delta translate to less severe disease.

We appreciate the insightful comment. In general, based on the vast literature and our observations, Omicron infection has been associated with a mild disease. However, we did not have access to non-hospitalized COVID-19 patients. This has been discussed as one of the study limitations (page 28 line 3-7).

Wuhan is misspelled repeatedly in Figures 2 and 4.
Corrected, thank you.

Reviewer #2 (Comments for the Author):

This study has a lot of interesting observations in it that are valuable due to overall good study design. I really appreciated that the authors timed the collections from patients to a standardized 2 weeks post start of symptoms or diagnosis of COVID, as that timing was quite appropriate for observations on the T cell compartment in particular. Also, having synced timing as well as having all patients on study being unvaccinated were critical for the observations as a whole.

We appreciate the vote of confidence.

It is not clear in the manuscript how the strain of COVID each patient on study had was determined - it should be stipulated whether the patient samples were subjected to strain-specific PCR tests, or if it was inferred based on timing of diagnosis relative to dominant strains at the time.

Thank you for the insightful comment, performing strain-specific PCR was not done on all patients but based on random PCR tests and timing of dominant strain at the time of diagnosis we determined the viral strain/variant. For clarification, we have included this statement in page 6 lines 4-7.

In Fig 1A, the PCA plot does not convincingly show a difference in the distance between HC and Wuhan strains versus patients with other strains as the authors describe in the text. There is a lot of spread in this data, so if that overall distance was determined mathematically and shows a significant difference, it should be shown, but otherwise I do not think that conclusion is valid.

Figures 1E-G do support that conclusion by demonstrating clear differences in the cytokine levels in patient blood in different groups, but the PCA plot does not add to/support that conclusion, so I would recommend removing Fig 1A, or at least removing the conclusion that there is a clear difference in COVID infected groups from each other in that analysis.

We appreciate the insightful suggestion. In agreement, we have modified the statement for clarification (Page 9 the last line and page 10 lines 1-3).

In Figure 3, the figure legend refers to Fig 3I as showing IL6 levels, but the graph itself has TNFalpha instead on the axis. The opposite is the case for 3J.

Thank you for identifying misplaced plots. This has been an error, which is corrected now.

There are a few concerning details in Figure 4 that need to be addressed. The PCA analysis done here includes as inputs both the galectin levels in the blood and the sCD14 and sCD163 data. It is not clear to me why comparisons including those particular data would be appropriate here, since Galectins are widely expressed. Galectin 3 is mainly derived from monocytes and mo-DCs, so that could be argued as related to the sCD14 measurements, but Gal 8 and Gal9 are widely expressed in non-immune cells.

Thank you for the insightful comment. This is correct that non-immune cells in addition to immune cells are a source of Gal-8 and Gal-9, however, these galectins can stimulate different immune cells. Gal-9 stimulates monocytes and NK cells and Gal-8 can also stimulate B cell and monocytes and etc. This has been further clarified (page 14 last lines and page 15 lines 1-4).

Furthermore, in the figures for the PCA analysis, in Fig 4 F, I believe that "ICU" was mistakenly substituted for "Wuhan", which is confusing.

Sorry for the mistake! It should be Wuhan and has been corrected.

Also, in the text of the manuscript, the authors indicate that the data suggests that infection with the Omicron variant is associated with "milder" activation of innate immune cells based on this PCA analysis, but this is not supported by the sCD14 data in Figure 4A, and in Fig 4B the sCD163 is lower than the Wuhan strain in both the Omicron patients and the Delta infected patients. Therefore, the differences seen in the PCA analysis seem to be driven largely by the inclusion of the Galectin data in that analysis, which is already detailed in Figure 2, and cannot be directly and solely tied to monocyte/macrophage activation.

Thank you for the suggestion. We have modified the text to reflect the reviewer's important point (page 14 lines 22-23, page 15 lines 1-4) and page 26 lines 9-12.

This is a smaller point as I do think the figures and patient information table are clear on this point, but in the text there are a number of places where, when data is similar between different groups, phrases like "patients infected with Delta and Wuhan strains" are used. I find this a bit unclear as it sounds like the patients are infected with both strains at once, rather than referring to the two groups having similar data readouts - I would recommend rephrasing to "patients infected with Delta or Wuhan strains" (for example) in these cases.

We appreciate the suggestion, we have modified the text accordingly.

July 13, 2023

Dr. Shokrollah Elahi
University of Alberta
Dentistry and Medical Microbiology and Immunology
11361 87th Ave NW
Edmonton, AB T6G2E1
Canada

Re: Spectrum01256-23R1 (Analysis of SARS-CoV-2 isolates namely the Wuhan strain, Delta and Omicron variants identifies differential immune profiles)

Dear Dr. Shokrollah Elahi:

Your manuscript has been accepted, and I am forwarding it to the ASM Journals Department for publication. You will be notified when your proofs are ready to be viewed.

Sincerely,

Leonidas Stamatatos
Editor, Microbiology Spectrum
